

# A deep stratosphere-to-troposphere ozone transport event over Europe simulated in CAMS global and regional forecast systems: Analysis and evaluation

Dimitris Akritidis[1], Eleni Katragkou[1], Prodromos Zanis[1], Ioannis Pytharoulis[1], Dimitris Melas[2], Johannes Flemming[3], Antje Inness[3], Hannah Clark[4], Matthieu Plu[5], and Henk Eskes[6]

[1]Department of Meteorology and Climatology, School of Geology, Aristotle University of Thessaloniki, Thessaloniki, Greece
[2]Laboratory of Atmospheric Physics, Physics Department, Aristotle University of Thessaloniki, Thessaloniki, Greece
[3]European Centre for Medium-Range Weather Forecasts, Reading, UK
[4]Laboratoire d'Aérologie, Univesíté de Toulouse, CNRS, UPS, France
[5]Centre National de Recherches Météorologiques, Météo-France-CNRS, UMR 3589, Toulouse, France
[6]Royal Netherlands Meteorological Institute (KNMI), De Bilt, the Netherlands
*Correspondence to:* D. Akritidis (dakritid@geo.auth.gr)

**Abstract.** Stratosphere-to-troposphere transport (STT) is the dominant natural source of tropospheric ozone, which can occasionally influence ground-level ozone concentrations relevant for air quality. Here, we analyse and evaluate the Copernicus Atmosphere Monitoring Service (CAMS) global and regional forecast systems during a deep STT event over Europe for the time period from 04 to 09 January 2017. The predominant synoptic condition is described by a deep upper level trough over

eastern and central Europe favouring the formation of tropopause folding events along the jet stream axis and therefore the intrusion of stratospheric ozone into the troposphere. Both global and regional CAMS forecast products reproduce the hook-shaped streamer of ozone-rich and dry air in the middle troposphere depicted from the observed satellite images of water vapor. The CAMS global model successfully reproduces the folding of the tropopause at various European sites, such as Trapani (Italy), where a deep folding down to 550 hPa is seen. The stratospheric ozone intrusions into the troposphere observed

by WOUDC ozonesonde and IAGOS aircraft measurements are satisfactorily forecasted up to three days in advance by CAMS global model in terms of both temporal and vertical features of ozone. The fractional gross error (FGE) of CAMS ozone Day-1 forecast between 300 and 500 hPa is 0.13 over Prague, while over Frankfurt is 0.04 and 0.19, highlighting the contribution of data assimilation which in most cases improves the model performance. Finally, the meteorological/chemical forcing of CAMS global forecast system in the CAMS regional forecast systems is found to be beneficial for predicting the enhanced

ozone concentrations in the middle troposphere during a deep STT event.

## 1   Introduction

Ozone is a key species in tropospheric chemistry, as it largely regulates the oxidation capacity of the troposphere (Monks, 2005). Excessive ozone concentrations near the earth's surface are known to be a risk for both public health and the ecosystems (WHO, 2003; Fuhrer et al., 1997). Moreover, tropospheric ozone is an important greenhouse gas (Solomon et al., 2007),



particularly in the upper troposphere due to its high radiative forcing efficiency (Lacis et al., 1990). Although photochemistry is the dominant source of tropospheric ozone (e.g. Crutzen, 1974; Fishman et al., 1979; Logan, 1985; Monks, 2000; Lelieveld and Dentener, 2000), the downward transport of ozone from the stratosphere is also an important process for the tropospheric ozone budget (e.g. Danielsen, 1968; Follows and Austin, 1992; Roelofs and Lelieveld, 1997; Stohl et al., 2003; Cristofanelli et al., 2006; Ordóñez et al., 2007; Zanis et al., 2014; Akritidis et al., 2016).

Deep and intense intrusions of stratospheric air penetrating down to lower tropospheric levels or even to the planetary boundary layer are more relevant than shallow ones for the atmospheric chemical composition, as they clearly lead to irreversible mixing of stratospheric and tropospheric air and hence to tropospheric composition changes affecting local air quality (Stohl et al., 2000; Cooper et al., 2005, 2011; Gerasopoulos et al., 2006; Akritidis et al., 2010; Cristofanelli et al., 2010; Ambrose et al., 2011; Lefohn et al., 2011, 2012; Emery et al., 2012; Langford et al., 2012; Lin et al., 2015; Knowland et al., 2017). Furthermore, recent modelling studies indicate that the role of stratosphere-to-troposphere transport (STT) to near surface ozone may be of even greater importance than anticipated in the 1990s and 2000s' (Zhang et al., 2011; Lin et al., 2012; Lefohn et al., 2014; Zanis et al., 2014).

Tropopause folds are considered as the main mechanism for STT events (Stohl et al., 2003). In principle, they are developed in the jet stream entrance, as a result of the ageostrophic flow, and are associated with penetrations of stratospheric air into the underlying troposphere (Danielsen and Mohnen, 1977) known as stratospheric intrusions. The key features of stratospheric intrusions are ozone-rich air, anomalously high potential vorticity (PV) levels and low water vapor mixing ratio (Holton et al., 1995; Wimmers et al., 2003). Following the transport into the troposphere, stratospheric air is quasi-adiabatically stirred by large-scale disturbances, which might result in the development of elongated streamers that can further dissipate down to smaller scales by non-conservative processes and lead to irreversible mixing with the surrounding air (Shapiro, 1980; Appenzeller and Davies, 1992; Forster and Wirth, 2000). In general, the vast majority of tropopause folds are of limited vertical extent and their spatio-temporal occurrence is mostly governed by both the position and the intensity of the subtropical jet stream (Holton et al., 1995; Stohl et al., 2003). Thus, the northern hemisphere tropopause folds frequency exhibits a maximum in the subtropics and during winter (Sprenger et al., 2003; Škerlak et al., 2015), while during summer a hotspot of tropopause fold activity is found over the eastern Mediterranean, Middle East and the Iran-Afghanistan regions, regulated by the complex interaction between the subtropical jet and the South Asian Monsoon anticyclone (Tyrlis et al., 2014). Deeper folds are also observed in the subtropics and further north over the North Atlantic storm track, most often during winter (Sprenger et al., 2003; Škerlak et al., 2015).

In the past, several studies have focused on the investigation of the prevailing synoptic and dynamic conditions governing the formation, evolution and intensity of tropopause folds and stratospheric intrusions (e.g. Shapiro, 1980; Appenzeller and Davies, 1992; Price and Vaughan, 1993; Lamarque and Hess, 1994; Vaughan et al., 1994; Wirth, 1995; Langford et al., 1996; Appenzeller et al., 1996; Baray et al., 2000; Forster and Wirth, 2000), while others explored the impact of tropopause folds on tropospheric ozone distribution and variability (e.g. Austin and Follows, 1991; Ancellet et al., 1994; Davies and Schuepbach, 1994; Beekmann et al., 1997; Bithell et al., 2000; Stohl et al., 2000; Zanis et al., 2003; Cooper et al., 2005; Cristofanelli et al., 2006; Trickl et al., 2010, 2011; Akritidis et al., 2016).



Copernicus is the European Union's Earth Observation program. The Copernicus Atmosphere Monitoring Service[1] (CAMS) is one of the six thematic areas that Copernicus addresses. CAMS uses a comprehensive global assimilation and forecasting system that estimates the state of the atmosphere and its composition on a daily basis, combining information from models and observations, providing daily 5-days forecasts of atmospheric composition fields, such as chemically reactive gases and

aerosols (Flemming et al., 2015; Inness et al., 2015). The CAMS global modelling system is also used to provide the boundary conditions for the CAMS ensemble of regional air quality models, which produce 4-day forecasts of European air quality. CAMS is in succession to the EU funded projects MACC (Monitoring Atmospheric Composition and Climate), and MACC-II (Interim Implementation) which were established to build and demonstrate a core capability for providing a comprehensive range of services related to the chemical and particulate composition of the atmosphere (Hollingsworth et al., 2008; Flemming

et al., 2009; Eskes et al., 2015).

The aim of this work is a process oriented analysis and evaluation of the CAMS global and regional forecast modelling systems for a deep STT event which affected tropospheric ozone in different parts of Europe. The added value of this work is the linkage between the global and regional services offered by CAMS, via the comparison of an ensemble of high-resolution forecast simulations by the CAMS regional air quality models with a forecast simulation by the global CAMS model in an

event of a deep STT. It also investigates whether representations of upper tropospheric dynamical/chemical processes in the CAMS global forecasting system are realistic and how adequately the global forcing can contribute to accurate regional air quality forecasts. This paper is structured in the following way. Section 2 describes the CAMS forecasting system and the observational validation data used in this study. Section 3 shows the results and Section 4 presents the main conclusions.

## 2   CAMS forecasting systems and observational data

**2.1   Composition in the ECMWF Integrated Forecasting System (IFS)**

The operational CAMS global forecasting system uses fully integrated chemistry in the European Centre for Medium-Range Weather Forecasts (ECMWF) Integrated Forecasting System (IFS). The IFS meteorology drives atmospheric composition changes and the IFS simulates atmospheric chemistry at a resolution of about 40 km (Flemming et al., 2015). CAMS uses the IFS data assimilation system to assimilate observations of atmospheric composition and includes weather-chemistry feedbacks.

Details of the ECMWF's 4D data assimilation system for aerosol, greenhouse gases and reactive gases can be found in Inness et al. (2015).

In addition to chemistry IFS also includes greenhouse gases (Engelen et al., 2009; Massart et al., 2016; Agusti-Panareda et al., 2017) and aerosols (Benedetti et al., 2009; Morcrette et al., 2009). IFS applies the Carbon Bond 2005 (CB05) chemical mechanism, which describes tropospheric chemistry with 55 species and 126 reactions (Flemming et al., 2015). Stratospheric

ozone chemistry in IFS is parameterized by the "Cariolle-scheme" (Cariolle and Déqué, 1986; Cariolle and Teyssedre, 2007). Chemical tendencies for stratospheric and tropospheric ozone are merged at an empirical interface of the diagnosed tropopause

---

[1]atmosphere.copernicus.eu





height in IFS (Flemming et al., 2015). In this paper we use IFS Day-1 forecasts of ozone, geopotential, u and v wind components, specific humidity and PV. In order to assess the impact of data assimilation on ozone representation during an STT event, an additional IFS control run without data assimilation (free running ozone) is used for intercomparison. Moreover, to evaluate the forecast performance of CAMS global forecast system during the STT event the IFS Day-2 to Day-5 forecasts of

ozone are also used.

## 2.2    CAMS Air Quality Regional Ensemble

The CAMS regional forecasting service is operated by Météo-France and provides daily 4-days forecasts of the main air pollutants and pollens, from seven state-of-the-art regional atmospheric chemistry models (http://atmosphere.copernicus.eu/ documentation-regional-systems) and from the median ensemble calculated from the 7 model forecasts. The 96h forecasts are

available with an hourly resolution and a spatial resolution of 0.1° from the surface up to 5 km. Currently the CAMS regional ensemble (RegEns) consists of the following regional models: CHIMERE from INERIS (National Institute for Industrial Environment and Risks) (Menut et al., 2014), EMEP from MET-Norway (Simpson et al., 2012), EURAD-IM from University of Cologne (Memmesheimer et al., 2004), LOTOS-EUROS from KNMI (Royal Netherlands Meteorological Institute) and TNO (Netherlands Organisation for Applied Scientific Research) (Schaap et al., 2008), MATCH from SMHI (Swedish Meteorolog-

ical and Hydrological Institute) (Robertson et al., 1999), MOCAGE from Météo-France (Guth et al., 2016) and SILAM from FMI (Finnish Meteorological Institute) (Sofiev et al., 2015). All regional model data are produced on a horizontal domain of 25°W-45°E and 30°N-70°N, covering a large European domain. The RegEns members have been documented and evaluated during the MACC projects (Marécal et al., 2015). The ozone results from RegEns and RegEns members presented here, correspond to Day-1 forecasts. The meteorological conditions in every model are driven by the operational ECMWF meteorological

forecasts, which are at 10 km horizontal resolution during the period of the study. The anthropogenic emissions are issued from the TNO MACC-III emission inventory over Europe for year 2011, which is an updated version of the TNO MACC-II inventory (Kuenen et al., 2014). All models use as lateral boundary conditions the concentrations of gas and aerosol species from the global CAMS system, which makes the regional model outputs consistent with the global model output. The differences between the seven models thus come from the different representation of the chemistry and aerosols, of the physical and

dynamical processes and of the natural emissions inside the domain.

## 2.3    Observational data

The observational data used in this paper include images by the Meteosat Second Generation (MSG) (Geo-Stationary) Satellite (NERC Satellite Receiving Station, Dundee University, Scotland, http://www.sat.dundee.ac.uk/) (last access: 17 March 2017). MSG carries the Spinning Enhanced Visible and InfraRed Imager (SEVIRI) instrument, which has the capacity to observe the

Earth in 12 spectral channels. Here, we present images from the MID-IR/Water Vapour channel (5.35-7.15 µm) for 12Z 06 January 2017 and 12Z 07 January 2017. Radiosonde data in the form of Skew-T Log-P diagrams (taken from the Wyoming University, Department of Atmospheric Science, http://weather.uwyo.edu/upperair/sounding.html) (last access: 27 April 2018) are used from four european stations:



(i) Norderney [10113], Germany, 53.71°N-7.15°E (12Z 03 January 2017 and 12Z 04 January 2017)

(ii) Muenchen-Oberschlssheim [10868], Germany, 48.25°N-11.55°E (00Z 04 January 2017 and 12Z 05 January 2017)

(iii) Trapani [16429], Italy, 37.91°N-12.50°E (00Z 05 January 2017 and 00Z 06 January 2017)

(iv) Heraklion [16754], Greece, 35.33°N-25.18°E (12Z 05 January 2017 and 00Z 08 January 2017)

Ozonesonde data over Prague [STN242], Czech-Repuplic (50.00°N-14.44°E) are obtained from the World Ozone and Ultra-violet Radiation Data Center (WOUDC) (WMO/GAW Ozone Monitoring Community) for 12Z 02 January 2017 and 12Z 04 January 2017 (last access: 09 June 2017).

Also used are aircraft ozone measurements from the IAGOS (In-service Aircraft for a Global Observing System) programme where instruments are carried on commercial airlines. In IAGOS CORE, instruments measure ozone, carbon monoxide and

water vapour along with meteorological parameters and cloud particles. Details of the IAGOS project can be found in Petzold et al. (2015), with the technical aspects of the instrumentation, operations and validation in Nédélec et al. (2015). Ozone and carbon monoxide are provided to CAMS in near real time for monitoring atmospheric composition. For the purposes of this validation in near real time, the data are provided after only an initial validation. After the instruments have been operating for a period of 6-12 months they are then calibrated in the laboratory and a final version of the data is released. The data used here

have therefore been validated but not yet calibrated. However, the ozone measurements are not expected to change significantly. Landing and take-off profiles are compared with the models at Frankfurt airport. It should be noted that the profiles are not strictly vertical. To this end and in order to perform a more realistic evaluation of CAMS models, according to the flight position (longitude, latitude, pressure) the respective grid points are extracted at the nearest time to that of the take-off or landing for both IFS and RegEns.

## 3   Results

### 3.1   Synoptic analysis

In early January 2017, severe winter weather struck several European regions, namely the Baltic Sea, northern Germany, Italy, the Balkan Peninsula and Turkey, with floodings, extreme cold and snow (Lentze, 2017). The international news media reported that at least 61 people died because of the extremely cold weather conditions in central, eastern and southern Europe

(Associated Press, 2017). The prevailing synoptic conditions associated with the these weather events are depicted in Figure 1, which presents the temporal evolution (every 12 hours) of IFS geopotential height, wind speed and wind direction at 300 hPa during the time period 03-09 January 2017. An upper-level ridge gradually formed over the eastern Atlantic and western Europe in conjunction with a deep upper-level trough over eastern and central Europe. Additionally, the jet stream was found on the western side of the upper level trough, with wind speeds occasionally exceeding $65 \, \mathrm{m \, s^{-1}}$ (12Z 04 January 2017 and

00Z 05 January 2017). This synoptic situation resulted in the advection of very cold arctic air-masses towards the eastern, central and southern Europe and favoured the formation of tropopause folds along the path of the jet stream. On its later stage



(00Z 08 January 2017 and after) the southernmost part of the system detached from the main stream, forming a cutoff low over the Balkans. The IFS temperatures at 850 hPa, averaged from 00Z 07 January to 21Z 10 January 2017, were below -14°C in most of the Balkans, reaching values below -18°C in western Balkans (not shown). To stress the exceptional intensity of the cold intrusion, it is noted that the monthly mean climatological temperatures for January at 850 hPa, derived from ERA-Interim

reanalyses for the 1981-2010 period, are not lower than -4°C in the Balkan region (not shown).

The horizontal thermal advection at 850 hPa was calculated at 3 hours intervals, using the IFS data and employing second order centered finite differences for the estimation of the horizontal derivatives. Cold advection at 850 hPa occurred in large parts of central, eastern and southern Europe in early January. Strong negative values of the horizontal thermal advection (< -1.5 K/hr) were exhibited continuously in large parts of Italy (05-07 January), northern Balkans and central Europe (04-09

January), western Balkans along the Adriatic coast (05-11 January) and northern Greece, southern FYROM and southwest Bulgaria (06-09 January). The latter maximum in cold advection resulted in a record period of 7 (5) consecutive days with frost (maximum daily temperature below 0°C) from 06 to 12 (07 to 11) January at Thessaloniki (northern Greece), which is located a few meters above the sea-level.

To further explore the meteorological conditions and to investigate the case of stratospheric intrusions into the troposphere

during the examined period, several stratospheric tracers are analyzed from both IFS and observations. The water vapour satellite images at 12Z on 06 and 07 January 2017 presented in Figure 2a and b, respectively, display a "hook-shaped" streamer of dry air (dark shades) extending from northeastern Europe to the central Mediterranean. This is a typical pattern encountered during STT events (Zanis et al., 2003; Gerasopoulos et al., 2006; Akritidis et al., 2010). The fields of IFS specific humidity at 500 hPa on the same days (Fig. 2c and d) resemble the observed satellite images. These depict a hook-shaped region of air

with low specific humidity, affirming that the presence of dry air into the troposphere is well captured by the IFS global model. The respective PV isosurfaces of 1.5 pvu (Fig. 2c and d) overlap the band of dry air in the troposphere, while high ozone concentrations, up to 130 ppb, are also found over this dry streamer (Fig. 2e and f). Altogether, Figure 2 indicates that this dry air with relatively high PV values and high ozone concentrations is of stratospheric origin.

### 3.2   Tropospheric ozone distribution in CAMS models

Figure 3 presents the evolution (12 hours interval) of ozone concentrations exceeding 50 ppb, geopotential height and PV isosurfaces of 1.5 pvu from IFS at 500 hPa for the time period 04-08 January 2017, to examine ozone enhanced in the middle troposphere owing to STT in relation to the predominant synoptic-dynamic conditions. On 12Z 04 January 2017 a streamer of high ozone concentrations with values up to about 100 ppb is found over Baltic Sea and northern Germany, near the ridge exit and trough entrance, where convergence and descending motions prevail, and in the vicinity of the jet stream (Fig. 1). During

the next 24 hours as the system moves further south the streamer of high ozone concentrations crosses central Europe following the path of the jet stream. On 00Z and 12Z 06 January 2017 ozone concentrations exceeding 130 ppb linked with high PV values (> 1.5 pvu), are found over the central Mediterranean, highlighting the vertical transport of ozone from the stratosphere down to the middle troposphere. During the next 48 hours, the high ozone streamer moves further eastward affecting the island of Crete (07 and 08 January 2017) and gradually dissipates.



In order to explore the capability of the regional models to reproduce the enhanced ozone seen in the mid-troposphere due to STT, the fields of RegEns ozone exceeding 50 ppb at 5000m are shown in Figure 4 for the same dates as in Figure 3. Visual inspection of Figure 4 indicates that the RegEns compares well with IFS as it synchronously captures the spatial distribution of ozone concentrations. In more detail, the hook-shaped patterns of high ozone are well seen in the CAMS regional product, with

ozone mixing ratios exceeding 90 ppb on 12Z 06 January 2017 over the central Mediterranean. Although the spatio-temporal features of ozone in the RegEns agree well with that of the IFS, in quantitative terms the regional product exhibits lower ozone concentrations compared with the global. This is likely due to the fact that (a) the RegEns is presented at 5000m level (the uppermost level available) and the IFS at 500 hPa, (b) different resolution and advection schemes are used in global and regional models and (c) pressure and temperature values from US Standard Atmosphere (USAF, 1976) were used for units

conversion in RegEns. Overall, the agreement between the CAMS global and regional products highlights the critical role that the IFS boundary conditions and meteorological drivers play in the regional models for forecasting an STT event and the induced downward transport of ozone.

### 3.3 Vertical structure and analysis of STT event

Four sites are selected (Norderney, Germany; Muenchen, Germany; Trapani, Italy; Heraklion, Greece), located within the

system transit path with available radiosonde observations, in order to study the vertical structure of the STT event and the subsequent transport of stratospheric ozone into the troposphere. To better depict the impact of STT on tropospheric ozone, two dates for analysis are selected for each site: one prior and one during the STT occurence.

Starting from Norderney (see location in Fig. 5a), the Skew-T Log-P diagrams for 12Z 03 January 2017 and 12Z 04 January 2017 are presented in Figure 5b and c, respectively. As can be seen from the comparison between the two figures a distinct de-

crease of humidity (departure of dewpoint curve (left) and temperature curve (right)) is found at 12Z 04 January 2017 between 250 and 400 hPa, while the tropopause drops to approximately 400 hPa. Furthermore, the vertical profile of the IFS ozone mixing ratio over Norderney during the examined dates (Fig. 5d) indicates a remarkable increase of ozone down to 400 hPa, verifying the aforementioned observed folding of the tropopause. A comprehensive view of the induced stratospheric intrusion over Norderney is provided through the longitude-pressure cross section at 53.6°N showing ozone, PV (2 pvu isosurface) and

wind speed at 12Z 04 January 2017 (Fig. 5e). An impressive downward penetration of ozone and PV (> 2 pvu) rich air down to approximately 600 hPa is found in the free troposphere and over the greater Norderney longitude band. The 2 pvu PV iso-surface (dynamical tropopause e.g., Hoskins et al. (1985)) illustrates the tropopause folding on the right side of the jet stream (black contours) and down to 450 hPa at 5°E. The stratospheric origin of ozone in the upper troposphere over Norderney is also supported by the IFS ozone and specific humidity time series at 400 hPa, revealing a significant anti-correlation at the

95% confidence level (Fig. 5f). The respective diagrams for Muenchen are presented in Figure 6 for 00Z 04 January 2017 and 12Z 05 January 2017. Similarly, an intrusion of dry air is observed in the upper and middle troposphere (down to 550 hPa) at 12Z 05 January 2017 (Figure 6b and c), which along with the sharp increase of IFS ozone above 550 hPa (Figure 6d), which is partially seen in RegEns vertical profiles, indicates the downward transport of dry stratospheric air into the troposhere. The longitude-pressure cross section over Muenchen at 12Z 05 January 2017 (Figure 6e) depicts the folding of the tropopause (2





pvu isosurface) in the vicinity of the jet stream and the associated vertical transport of ozone-rich air down to 600 hPa. In support of the above, the distinct increase of IFS ozone at 400hPa is combined with a sharp decrease of IFS specific humidity (significant anti-correlation at the 95% confidence level) (Fig. 6f).

Twelve hours later (00Z 06 January 2017), and as the system moved further south, a dramatic decrease of humidity is ob-
served in the middle troposphere and down to approximately 550 hPa over Trapani (Figure 7b and c), with specific and relative humidity at 500 hPa dropping from 0.75 $\mathrm{g\,kg^{-1}}$ and 58% (00Z 05 January) to 0.01 $\mathrm{g\,kg^{-1}}$ and 2% (00Z 06 January 2017) respectively. The IFS specific humidity values at 500 hPa for the same dates are 0.49 $\mathrm{g\,kg^{-1}}$ and 0.025 $\mathrm{g\,kg^{-1}}$, respectively, indicating that the sharp decrease of humidity is well reproduced by the CAMS global model. The IFS system captures the dynamical features of the stratospheric intrusion as it is depicted in the vertical profiles of ozone showing increased concen-
trations at 00Z 06 January 2017 down to 600 hPa, which is also seen in CAMS RegEns (Fig. 7d). The intense tropopause folding over Trapani is illustrated in Figure 7e with the dynamical tropopause dropping down to 550 hPa and ozone-rich air penetrating down to 800hPa. Again, a significant anti-correlation at the 95% confidence level is found between the IFS ozone and specific humidity time series at 400 hPa, indicating that the ozone increase results from the downward transport of ozone from the stratosphere (Fig. 7f). The three-dimensional field of IFS ozone concentrations exceeding 80 ppb at 00Z 06 January
2017 is presented in Figure 8a, depicting the stratospheric ozone intrusion into the troposphere and over the broader Trapani region. The three-dimensional IFS ozone concentration isosurface of 100 ppb (Fig. 8b, mind the angle of view) resembles the folding of the tropopause along a north-east oriented conceivable axis which concides with the high wind speed flow in the upper troposphere (Fig. 1). Later on and over Heraklion (see location in Fig. 9a), the Skew-T Log-P diagrams for 12Z 05 January 2017 and 00Z 08 January 2017 (Fig. 9b and c) and the respective vertical profiles of IFS ozone (Fig. 9d) reveal
the presence of dry ozone-rich air in the upper and middle troposphere (down to 500 hPa). The increase in IFS ozone time series at 400 hPa is synchronised with the decrease of IFS specific humidity (significant anti-correlation at the 95% confidence level), indicating that dry stratospheric air rich in ozone is transported into the troposphere over Heraklion (Fig. 9f). A more illustrative representation of the development and evolution of the examined STT event is provided in the three-dimensional animation (from 12Z 03 January 2017 to 21Z 08 January 2017 with 3 hours interval) of IFS ozone concentrations exceeding
80 ppb, in the Supplement.

### 3.4 Comparison with profile observations

In order to evaluate the forecasting capability of both IFS and RegEns regarding the downward transport of ozone during the examined STT event, we compare CAMS forecasts with profile observations from ozonesondes (WOUDC) and aircraft measurements (IAGOS). Two sites located across the passage of the examined system with availiable observational data during
the examined period were selected: a) Prague (ozonesondes) and b) Frankfurt (aircraft measurements). The model error is quantified using the fractional gross error (FGE) which ranges between 0 and 2, and behaves symmetrically with respect to under- and overestimation:

$$FGE = \frac{2}{N}\sum_{i}^{N}\left|\frac{M_i - O_i}{M_i + O_i}\right| \tag{1}$$



where $M_i$ represents the model value for level i, $O_i$ is the corresponding observed value and N is the number of sample values.

Figure 10 displays the vertical profiles of observed and forecasted (IFS and RegEns) ozone concentrations over Prague at 11Z (12Z for CAMS models) 02 January 2017 (prior the STT event) and 11Z (12Z for CAMS models) 04 January 2017 (during the STT event). The intercomparison between the observed vertical profiles of ozone on the two dates indicates a distinct increase

of ozone concentrations in the upper troposphere probably related to the vertical transport of ozone from the stratosphere, reaching down to approximately 500 hPa. Although the CAMS global model seems to underestimate (overestimate) ozone in (above) the free troposphere, the transition from the neutral condition to the STT event is well captured by IFS Day-1 forecast (Fig. 10), with an FGE value of 0.13 (300-500 hPa) on 04 January 2017. Whilst data assimilation resulted in overestimating ozone near the tropopause compared with the control run (Fig. 10a), it is clearly beneficial in reproducing the increase of

ozone in the upper troposphere during the STT event (Fig. 10b). Notably, the respective FGE value for the control run at 04 January 2017 is 0.29, revealing an improvement in model performance due to data assimilation. Ozone in the RegEns forecast is higher within the planetary boundary layer than in IFS, with a relatively small spread among the RegEns members (Fig. 10a). In the free troposphere, the range of regional variability increases, however the RegEns remains close to the global forecast. The RegEns is also able to reproduce the ozone enhancement, following closely the IFS forecast (Fig. 10b). Day-1 to Day-5

forecasts of IFS ozone indicate that the observed ozone increase in upper troposphere during the STT event is satisfactorily forecasted up to three days in advance with FGE values not higher than 0.22 (Fig. 10b).

Three ozone profiles from aircraft measurements (two take-offs and one landing) over the broader region of Frankfurt at 13Z 04 January 2017, 06Z 05 January 2017 and 13Z 05 January 2017 are compared with the respective IFS and RegEns ozone profiles in Figure 11. At 13Z (12Z for CAMS models) 04 January 2017 the profile of IFS ozone Day-1 forecast is found to be

in very good agreement with the IAGOS data, both depicting the increase of ozone down to approximately 500 hPa (Fig. 11a). The FGE was 0.04 for the 300-500 hPa altitude range. The respective profiles 17 hours later (06Z 05 January 2017) also reveal enhanced ozone concentrations in the upper troposphere, which are captured by IFS Day-1 forecast (Fig. 11b) (FGE=0.19 at 300-500 hPa). Finally, at 13Z (12Z for CAMS models) 06 January 2017, IFS Day-1 forecast is found to overestimate the observed high ozone concentrations between 250 and 350 hPa, while it qualitatively captures the observed high ozone

pattern in the middle troposphere between 400 and 600 hPa (Fig. 11c) (FGE=0.30 at 400-600 hPa). The advantageous role of data assimilation can be affirmed from the intercomparison with the IFS control run which exhibits FGE values of 0.34, 0.30 and 0.12 for the three dates respectively. A better agreement with observations is found for IFS when implementing data assimilation at 13Z 04 January 2017 and 06Z 05 January 2017 (Fig. 10a and b), while at 13Z 05 January 2017 (Fig. 10c) although the control run performs better in terms of bias the data assimilation seems to help in the direction of reproducing the

observed ozone peak in the middle troposphere. As concerns the RegEns, due to its limited vertical profile, up to about 550 hPa, the evaluation of its forecast performance is restricted. Nevertheless, there is a clear signal of increased ozone in the uppermost vertical level during all three dates. Regarding the forecast performance of CAMS global model a relatively good agreement with observations is seen up to forecast Day-3 at 13Z 04 January 2017 and 06Z 05 January 2017 (FGE values not higher than 0.25) (Fig. 11a and b), while at 13Z 05 January 2017 the observed ozone peak in the middle troposphere is somehow captured

up to forecast Day-3 but oversetimated. Figure 12 depicts the FGE values of IFS ozone in relation to the forecast day for



the observational instances of Prague and Frankfurt. Overall, a satisfactory forecast performance is revealed up to three days in advance with FGE values not higher than 0.3. Forecast Day-1 exhibits the best agreement with observations, while after forecast Day-3 more discrepancies are found between the forecast and the observations (see also Fig.10 and Fig.11).

## 4 Conclusions

We examined a deep STT event over Europe during the time period from 04 to 09 January 2017 in the CAMS global and regional forecast systems, assessing their capability to reproduce several key meteorological and chemical features of the event, with the aid of radiosonde, ozonesonde and aircraft observational data. The main results of the current study can be summarized as follows:

- A deep upper level trough extending over central Europe favoured the development of tropopause folds and subsequently
STT events along the jet stream axis at the west flank of the trough between 04 and 09 January 2017.

- The hook-shaped streamer of dry stratospheric air in the middle troposphere seen in water vapor satellite images is well reproduced by the CAMS forecast systems, with tongues of anomalously high ozone concentrations in both CAMS global and regional models.

- The observed (radiosondes) folding of the tropopause over various European sites is accurately reproduced by the CAMS
global model. The vertical profiles and cross sections of IFS ozone and PV indicate that the vertical extent of the observed tropopause drop is well captured at all four of the sites studies.

- The CAMS global system is found to be capable of capturing the evolution and vertical characteristics of the observed ozone field over Prague during the STT event. The observed ozone increase in the upper troposphere due to the strato-spheric ozone downward transport is relatively well captured by the IFS. In addition, the global CAMS ozone forecasts
in the greater Frankfurt area reveal an enhancement of ozone concentrations in the upper and middle troposphere as a result of the STT, which is in good agreement with the ozone measured by IAGOS aircraft.

- The evaluation of IFS ozone forecasts indicates that CAMS global system is capable of forecasting the enhanced ozone concentrations during the STT event over Prague and Frankfurt up to three days in advance, both qualitatively and quantitatively.

- Figure 10, Figure 11 and Figure 12 show that the use of data assimilation in the IFS is generally beneficial in forecasting the vertical and temporal variability of ozone during the examined STT event. Nevertheless, there are still discrepancies from the observations near the tropopause region as the sharp gradients around the tropopause are difficult to capture in global models (Clark et al., 2007; Gaudel et al., 2015).

- Despite the limited vertical profile of RegEns forecast data, the CAMS regional models show an increase of ozone in the
uppermost level for all instances where the STT reached or exceeded that level.




Overall, this process-oriented analysis and evaluation study indicates that the CAMS global and regional forecast modelling systems are able to capture the specific regional meteorological and air quality characteristics of a specific deep STT event over Europe in January 2017. It also highlights the importance of data assimilation in the CAMS global model as well as of the meteorological/chemical forcing to the CAMS regional forecast systems.

5 *Acknowledgements.* This work is performed within the framework of the service element "CAMS_84: Global and regional a posteriori validation, including focus on the Arctic and Mediterranean areas" of the Copernicus Atmospheric Monitoring Services (CAMS). ECMWF is the operator of CAMS on behalf of the European Union (Delegation Agreement signed on 11/11/2014). The CAMS_84 work is financially supported by ECMWF via its main contractor Royal Netherlands Meteorological Institute KNMI. The authors acknowledge the strong support of the European Commission, Airbus, and the Airlines (Lufthansa, Air-France, Austrian, Air Namibia, Cathay Pacific, Iberia, China 10 Airlines, Hawaiian Airlines so far) who carry the MOZAIC or IAGOS equipment and perform the maintenance since 1994. In its last 10 years of operation, MOZAIC has been funded by INSU-CNRS (France), Météo-France, Université Paul Sabatier (Toulouse, France) and Research Center Jülich (FZJ, Jülich, Germany). IAGOS has been additionally funded by the EU projects IAGOS-DS and IAGOS-ERI. The MOZAIC-IAGOS database is supported by AERIS (CNES and INSU-CNRS). Data are also available via AERIS web site www.aeris-data.fr. The authors acknowledge the use of Copernicus Atmosphere Monitoring Service Information [2017]. We also acknowledge the WOUDC, 15 the Department of Atmospheric Science of the Wyoming University and the NERC Satellite Receiving Station of the Dundee University for the free use of ozonesondes data, radiosondes data and satellite images respectively. Finally, the authors would like to acknowledge the free use of Python (www.python.org), Ferret (http://ferret.pmel.noaa.gov/Ferret/) and Mayavi (Ramachandran and Varoquaux, 2011) softwares for the analysis and graphics of the paper.



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

**Figure 1.** IFS geopotential height (in gpm; contours), wind speed (in m s$^{-1}$; color shaded) and wind direction (vectors) at 300 hPa, during the period 12Z 03 Jan 2017 to 00Z 09 Jan 2017 (12 hours interval).





**Figure 2.** Meteosat water vapor (5.35-7.15 µm) satellite images (a and b), IFS specific humidity (in g kg$^{-1}$; color shaded) and PV (1.5 pvu; contours) at 500 hPa (c and d), and IFS ozone mixing ratio (in ppb; color shaded) at 500 hPa at 12Z 06 Jan 2017 and 12Z 07 Jan 2017 respectively. Satellite images source: NERC Satellite Receiving Station, Dundee University, Scotland, http://www.sat.dundee.ac.uk/.





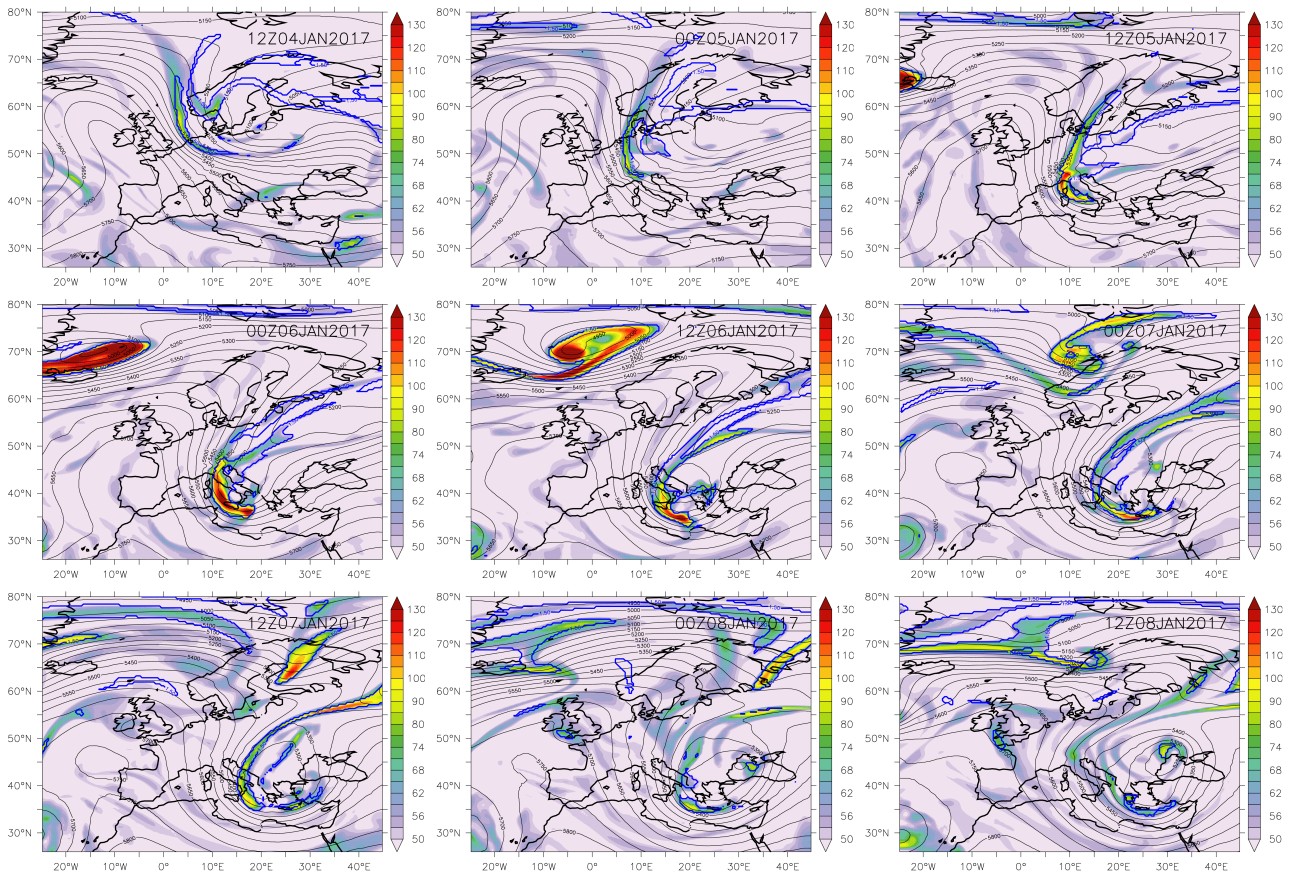

**Figure 3.** IFS ozone mixing ratio (in ppb; color shaded), geopotential height (in gpm, black contours) and PV (1.5 pvu; blue contours) at 500 hPa during the period 12Z 04 Jan 2017 to 12Z 08 Jan 2017 (12 hours interval).





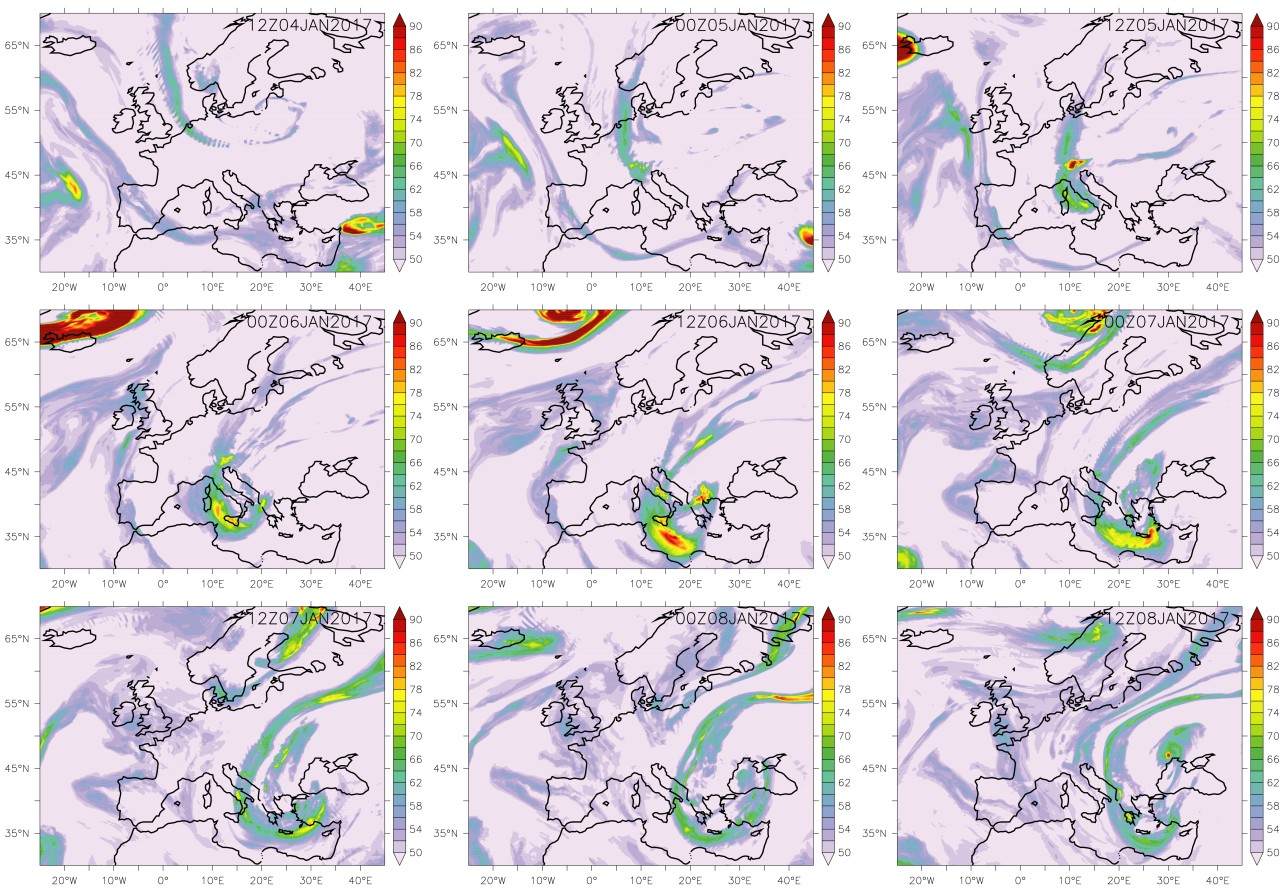

**Figure 4.** RegEns ozone mixing ratio (in ppb; color shaded) at 5000 m during the period 12Z 04 Jan 2017 to 12Z 08 Jan 2017 (12 hours interval).



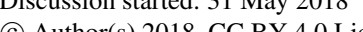



**Figure 5.** (a) Norderney, Germany location. (b) Skew-T Log-P diagrams at 12Z 03 Jan 2017 and (c) 12Z 04 Jan 2017. (d) Vertical profiles of IFS (blue) and RegEns (red) ozone mixing ratio (ppb) at 12Z 03 Jan 2017 (solid line) and 12Z 04 Jan 2017 (dashed line). The red bars denote the standard deviation among the regional emsemble members. (e) Longitude-pressure vertical cross-section at 53.6°N of IFS ozone mixing ratio (in ppb; color shaded), wind speed (in $\mathrm{m\,s}^{-1}$; black contours) and PV (2 pvu; blue contours) at 12Z 04 Jan 2017. (f) IFS ozone (blue) and specific humidity (orange) time series at 400 hPa.





**Figure 6.** (a) Muenchen, Germany location. (b) Skew-T Log-P diagrams at 00Z 04 Jan 2017 and (c) 12Z 05 Jan 2017. (d) Vertical profiles of IFS (blue) and RegEns (red) ozone mixing ratio (ppb) at 00Z 04 Jan 2017 (solid line) and 12Z 05 Jan 2017 (dashed line). The red bars denote the standard deviation among the regional emsemble members. (e) Longitude-pressure vertical cross-section at 48.4°N of IFS ozone mixing ratio (in ppb; color shaded), wind speed (in m s$^{-1}$; black contours) and PV (2 pvu; blue contours) at 12Z 05 Jan 2017. (f) IFS ozone (blue) and specific humidity (orange) time series at 400 hPa.





**Figure 7.** (a) Trapani, Italy location. (b) Skew-T Log-P diagrams at 00Z 05 Jan 2017 and (c) 00Z 06 Jan 2017. (d) Vertical profiles of IFS (blue) and RegEns (red) ozone mixing ratio (ppb) at 00Z 05 Jan 2017 (solid line) and 00Z 06 Jan 2017 (dashed line). The red bars denote the standard deviation among the regional emsemble members. (e) Longitude-pressure vertical cross-section at 38°N of IFS ozone mixing ratio (in ppb; color shaded), wind speed (in $\mathrm{m\,s^{-1}}$; black contours) and PV (2 pvu; blue contours) at 00Z 06 Jan 2017. (f) IFS ozone (blue) and specific humidity (orange) time series at 400 hPa.



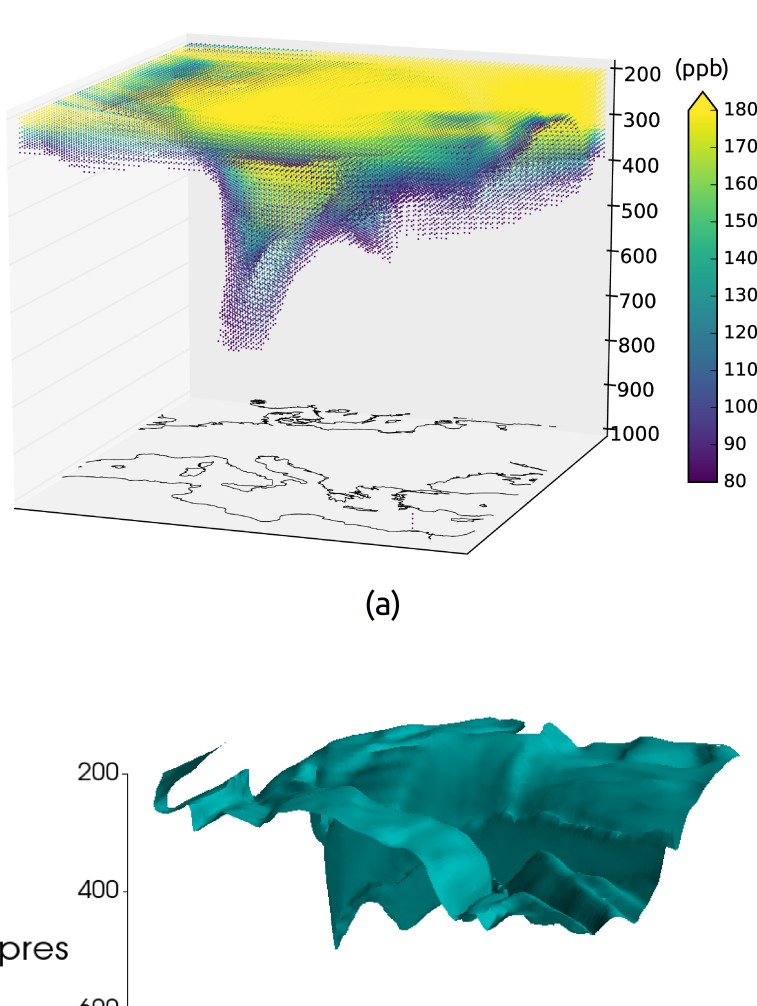

**Figure 8.** (a) Three-dimensional (longitude, latitude, pressure (hPa)) spatial distibution of IFS ozone concentrations exceeding 80 ppb at 00Z 06 January 2017. (b) Three-dimensional (longitude, latitude, pressure (hPa)) IFS ozone concentrations iso-surface of 100 ppb at 00Z 06 January 2017.





**Figure 9.** (a) Heraklion, Greece location. (b) Skew-T Log-P diagrams at 12Z 05 Jan 2017 and (c) 00Z 08 Jan 2017. (d) Vertical profiles of IFS (blue) and RegEns (red) ozone mixing ratio (ppb) at 12Z 05 Jan 2017 (solid line) and 00Z 08 Jan 2017 (dashed line). The red bars denote the standard deviation among the regional emsemble members. (e) Longitude-pressure vertical cross-section at 25.2°E of IFS ozone mixing ratio (in ppb; color shaded), wind speed (in m s⁻¹; black contours) and PV (2 pvu; blue contours) at 00Z 08 Jan 2017. (f) IFS ozone (blue) and specific humidity (orange) time series at 400 hPa.



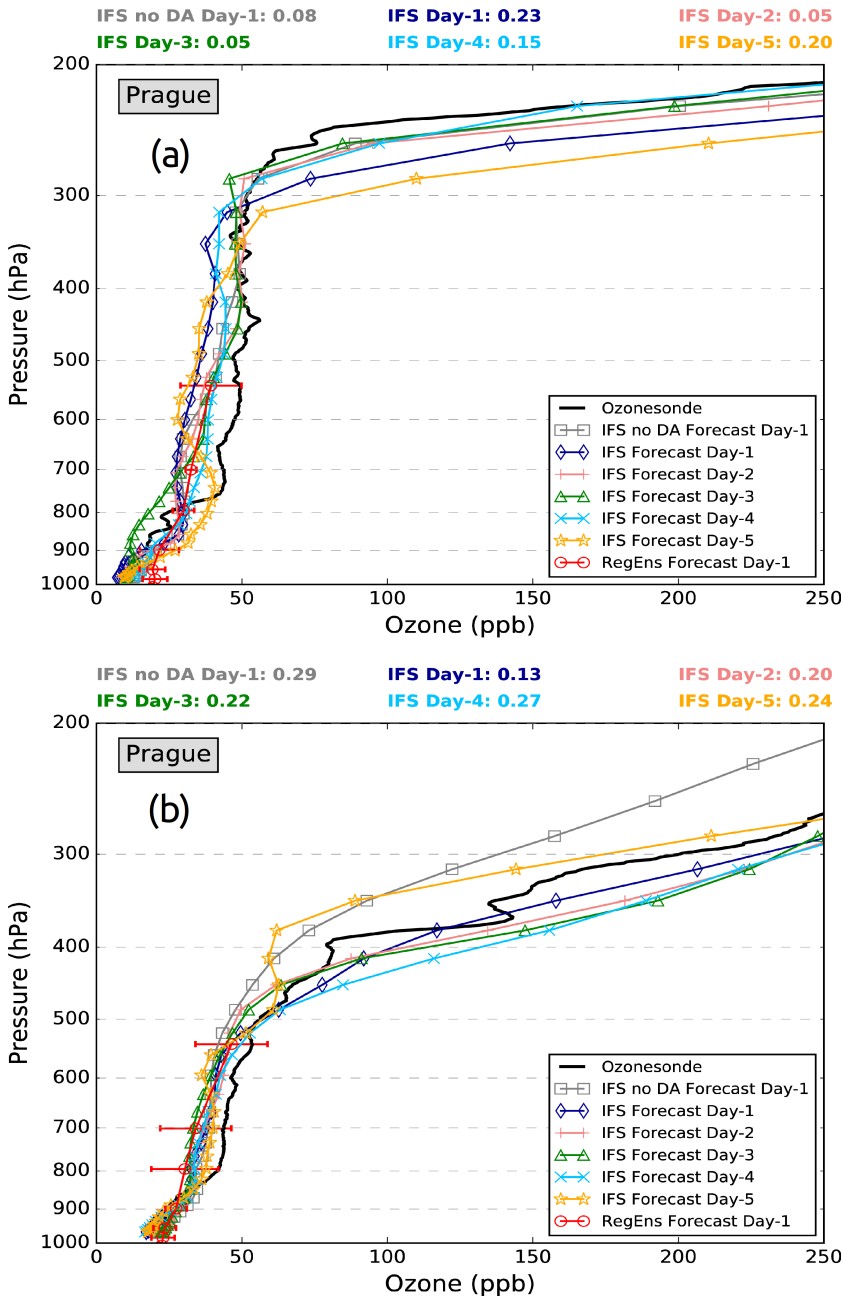

**Figure 10.** Vertical profiles of ozone mixing ratio (ppb) over Prague, Czech Republic (14.44°E, 50°N) for ozonesondes (black line), IFS forecast Day-1 (dark blue line), IFS forecast Day-2 (coral line), IFS forecast Day-3 (green line), IFS forecast Day-4 (light blue line), IFS forecast Day-5 (orange line), IFS no DA (without data assimilation) forecast Day-1 (grey line) and RegEns (red line) at (a) 12Z 02 Jan 2017 and (b) 12Z 04 Jan 2017. The red bars denote the standard deviation among the regional emsemble members. The numbers above the diagrams show the FGE values of IFS ozone (with the corresponding color) at 300-500 hPa.



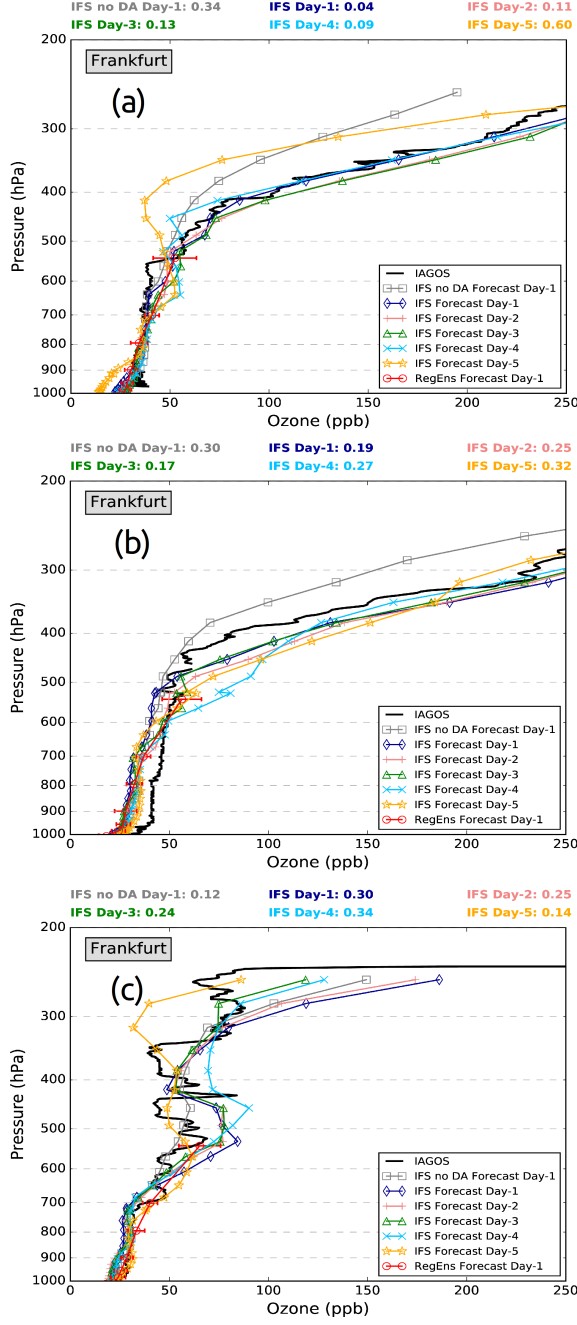

**Figure 11.** Profiles of ozone mixing ratio (ppb) over the broader area of Frankfurt (8.5°E, 50°E) for IAGOS (black line), IFS forecast Day-1 (dark blue line), IFS forecast Day-2 (coral line), IFS forecast Day-3 (green line), IFS forecast Day-4 (light blue line), IFS forecast Day-5 (orange line), IFS no DA (without data assimilation) forecast Day-1 (grey line) and RegEns (red line) during (a) 13Z 04 Jan 2017, (b) 06Z 05 Jan 2017, (c) 13Z 05 Jan 2017. The red bars denote the standard deviation among the regional emsemble members. The numbers above the diagrams show the FGE values of IFS ozone (with the corresponding color) at 300-500 hPa (a and b) and 400-600 hPa (c).



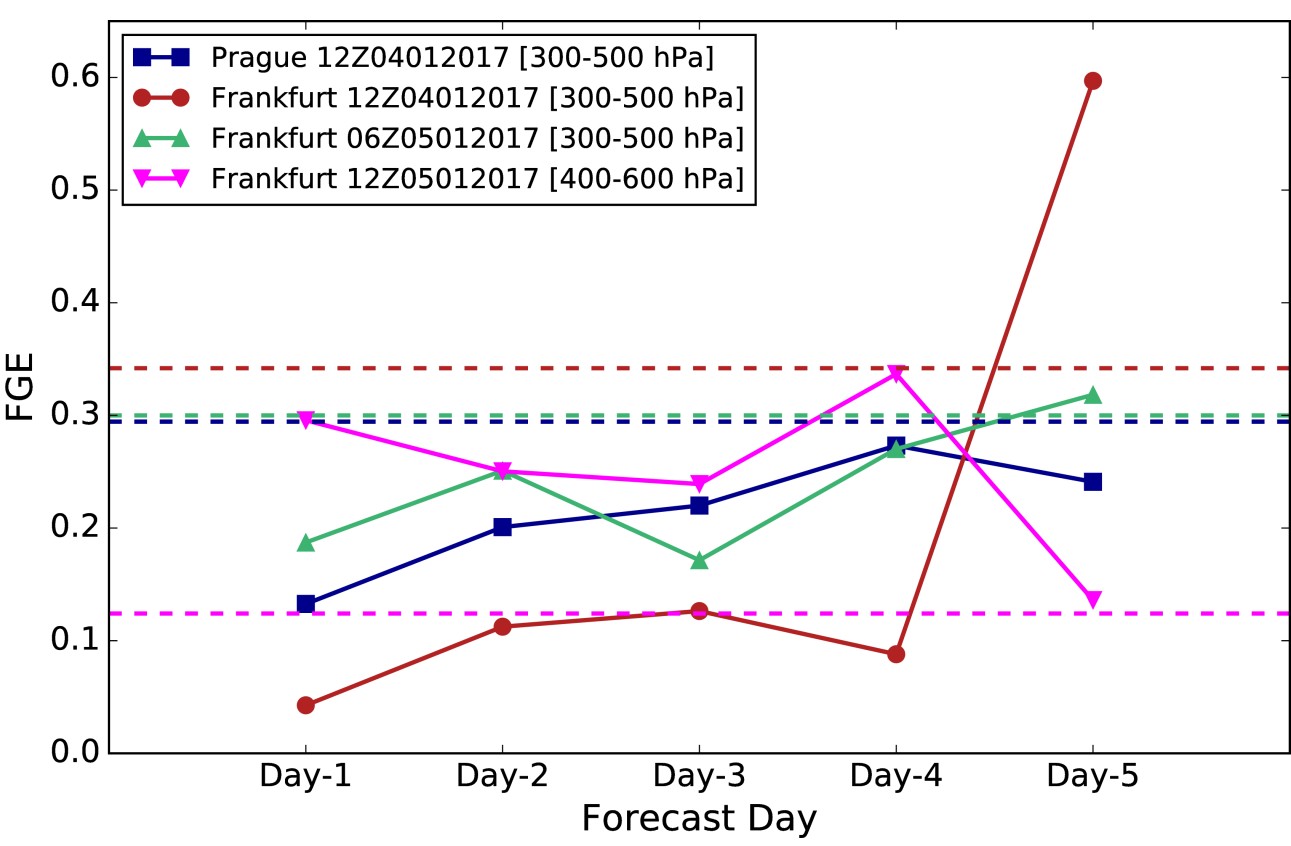

**Figure 12.** FGE values of IFS ozone for forecast days 1-5 over Prague (12Z 04 Jan 2017) and Frankfurt (12Z 04 Jan 2017, 06Z 05 Jan 2017 and 12Z 05 Jan 2017). The dashed colored horizontal lines represent the FGE values of IFS no DA ozone for forecast Day-1.