# Peer review of "A deep stratosphere-to-troposphere ozone transport event over Europe simulated in CAMS global and regional forecast systems: Analysis and evaluation"

_Atmospheric Chemistry and Physics, 2018_

## Referee Comment (RC1) · Anonymous Referee #1 · 16 Jul 2018

The review on "A deep stratosphere-to-troposphere ozone transport event over Europe simulated in CAMS global and regional forecast systems: Analysis and evaluation" by Akritidis et al.

General

This manuscript describes a deep stratosphere-to-troposphere event over Europe in 2017 winter that has been well captured by CAMS global and regional forecast models. The authors illustrated the simulated winds, geopotential height, PV, water vapour and ozone during the event. They compared the simulations with satellite data of water vapour , radiosonde, ozonesonde and aircraft observations. By putting all of the simulated and observed meteorological and chemical data together, the authors depicted the evolution of this event in detail and showed strong performance of the CAMS global and regional models.

Overall, this study is well conducted and has contributed to enhancing our knowledge of ozone transport from the stratosphere to troposphere. The presentation is overall clear. However, I have the following points for the authors to consider when revising their paper.

1. While the CAMS showed strong performance in capturing the stratospheric intrusion event, it is not clear (1) what are key schemes in the models that are responsible for the performance and (2) what advance this study has made comparing with earlier studies. Can the authors provide some assessments on the model prediction of ozone intrusion events? Do the models tend to overestimate or underestimate occurrence of these events?
2. There are more than one ozonesonde stations in Europe. The authors are encouraged to take advantage of the ozonesonde data from more ozonesonde stations to validate the model performance. In these validation, such as those in Figure 10, humidity can also be validated so to provide additional confidence on the model performance.
3. More description is required on the data assimilation. What kinds of observation data were used in the assimilation? Were the ozonesonde data at Prague or aircraft data at Frankfurt used? If so, this should be pointed out when discussing Figures 10 and 11.

Specific

1. Please indicate the locations of radiosond, ozonesonde stations, and Frankfun in Figures 1-4.
2. Figure 10, humidity data usually are available together with the ozonesonde data. Humidity can be validated at the same time.
3. References:
   P12, L3, CO 2 and CH 4?
   P12, L13, more information is required.
   P12,34, CATHALA?

P15, L22, Spell out the full name of the journal.

---

## Referee Comment (RC2) · Anonymous Referee #2 · 31 Jul 2018

Review of Akritidis et al

Akritidis et al present a study which assesses a stratosphere-to-troposphere transport event (STT) that occurred over Europe during the cold winter of 2017. STT is a very important source of ozone into the troposphere but has and remains challenging to simulate given the laminar like structures that are associated with these events and their transient in time nature. Arkitidis use results from a range of models including the ECMWF Copernicus Atmosphere Monitoring Service to understand the drivers for this event and to use the observations of this event from aircraft and ozonesondes to evaluate the models.

In general this is a well written manuscript and one I would recommend published after the following general and technical points are considered.

General comments:
1. A table with the model acronyms and set ups used in the analysis would be very useful for the reader. Most of the information is already in the text but I feel a table would help the reader quickly appreciate the differences between RegEns, CAMS and IFS.
2. What more can we learn from this event? The RegEns models all differ in structure and I wonder what further analysis could be done to help understand (a) the role of resolution in the vertical (b) the role of horizontal resolution in biases that occur in the models. The use of the IFS with and without DA is instructive but I feel there is more to be teased out from RegEns and would like to see some more effort to that extent.

Technical comments:
Page 1, line 1: Technical point. I'm not sure I agree with the opening statement. STT tends to produce about 200 Tg (O3)/yr. Lightning NOx (natural) produces XXX..
Page 3, line 24: I find the phrase "weather-chemistry feedback" a bit puzzling.. I think you could be more specific here. What exact feedbacks are included and how are they represented?
Page 4, line 2: I guess when you mean data assimilation you mean chemical data assimilation?
Page 4, line 8-9: Pernickety, I know, but you have used "seven" on line 8 and "7" on line 9 to refer to the number of models in the CAMS ensemble. Sticking with one or the other would be better.
Page 6, line 10: Define FYROM please.
Page 6, line 13: Remove "the" before sea-level.
Page 7, line 7: Can the authors confirm why pressure-interpolation of the chemical fields from model levels onto pressure levels could not be performed? I would think this is a fairly standard procedure. Could they elaborate on the errors introduced for example by not accounting for the exceptionally low real temperatures when using the US Standard Atmosphere for unit conversion? In addition, with respect to Figure 4, I would be intrigued to know what the spread is within RegEns or the standard deviation of the ensemble? In general, what can we learn more about the models from this event?
Page 8, line 16: I'm not sure what you mean by "mind the angle of view"?
Page 9, line 14: Can you speculate why the spread in RegEns increases in the vertical?

Page 10, line 1: What constitutes "satisfactory"?
Page 10, line 22: Insert "the" before CAMS.

Figure 2 caption, missing information about panels e and f.

---

## Author Comment (AC1) · 30 Sep 2018

Note: Reviewer's comments are presented in black font; authors' responses are presented in blue plain font; manuscript text quotations are presented in blue italics font.

Anonymous Referee #1

We would like to thank Reviewer #1 for his/her time devoted and the constructive and helpful comments.

General

This manuscript describes a deep stratosphere-to-troposphere event over Europe in 2017 winter that has been well captured by CAMS global and regional forecast models. The authors illustrated the simulated winds, geopotential height, PV, water vapour and ozone during the event. They compared the simulations with satellite data of water vapour , radiosonde, ozonesonde and aircraft observations. By putting all of the simulated and observed meteorological and chemical data together, the authors depicted the evolution of this event in detail and showed strong performance of the CAMS global and regional models.

Overall, this study is well conducted and has contributed to enhancing our knowledge of ozone transport from the stratosphere to troposphere. The presentation is overall clear. However, I have the following points for the authors to consider when revising their paper.

We thank the Reviewer for the comments, to which we will respond point by point.

1. While the CAMS showed strong performance in capturing the stratospheric intrusion event, it is not clear (1) what are key schemes in the models that are responsible for the performance and (2) what advance this study has made comparing with earlier studies. Can the authors provide some assessments on the model prediction of ozone intrusion events? Do the models tend to overestimate or underestimate occurrence of these events?

(1) The simulation of the intrusion is mainly driven by model dynamics. An important aspect is also the vertical model resolution; increasing the number of vertical layers, theoretically would improve the forecast evaluation metrics. (2) This evaluation work is the first in its kind, since the IFS system has not been evaluated before for stratospheric intrusions. It is a process oriented evaluation study which is complementary to the standard evaluation work performed within the CAMS84 service. This work is also original because we compare the global (IFS) with the regional European air quality forecasts, the latter driven by IFS. Therefore we give insight into a comparative model performance, and present the range of uncertainty in the regional ensemble.

As mentioned above this is the first study about the IFS performance for stratospheric intrusions, so there is not yet a systematic study on the IFS performance for such events. Although this is out of the scope of the current work, it is quite an interesting aspect for future work.

2. There are more than one ozonesonde stations in Europe. The authors are encouraged to take advantage of the ozonesonde data from more ozonesonde stations to validate the model performance. In these validation, such as those in Figure 10, humidity can also be validated so to provide additional confidence on the model performance.

We agree with the Reviewer's suggestion to include also humidity in the vertical profiles of ozone to increase confidence on the model performance. Apart from Figure 10 (Prague), we extended this suggestion also for Figures 5, 6, 7 and 9 (Norderney, Muenchen, Trapani and Heraklion) including relative humidity as a stratospheric tracer. The following figures present the updated vertical profiles including the observed and ifs relative humidity:

[Figure]

Figure 10a                    Figure 10b

Figure 5                       Figure 6

[Figure]

Figure 7                                    Figure 9

Please find the updated Figures 5, 6, 7, 9, 10 and captions in the revised manuscript: Figure 5: page 24; Figure 6: page 25; Figure 7: page 26; Figure 9: page 28; Figure 10; page 29.

Moreover, a small discussion on the relative humidity is included/modified in the revised manuscript:

Page 4, line 7: *"…specific humidity, relative humidity and PV..."*

Page 8, lines 6-8: *"The vertical profiles of the observed and IFS relative humidity (Fig. 5d) show a sharp decrease at 400 hPa, revealing that the intrusion of dry stratospheric air in the troposphere is well captured by the IFS.".*

Page 8, lines 17-18: *"…which along with the sharp increase/decrease of IFS ozone/relative humidity above 550 hPa (Figure 6d), which is partially seen in RegEns ozone vertical profiles, indicates the downward transport of dry stratospheric air into the tropopshere".*

Page 8, lines 27-28: *"On top of that, the vertical profiles of the observed and IFS relative humidity (Fig. 7d) indicate that the sharp decrease of humidity is well reproduced by the CAMS global model.".*

Page 9, lines 3-4: *"..and the respective vertical profiles of IFS ozone and relative humidity (Fig. 9d) reveal.."*

Page 9, lines 23-24: *"In support of the above findings, the respective vertical profiles of the observed and IFS relative humidity show both a distinct decrease at 500 hPa."*

The Reviewer suggests using additional ozonesonde stations in our analysis. We agree that there are available ozonesonde data from other European stations. Nevertheless, as our study is strictly focused on stratosphere-to-troposphere transport (STT), we are only interested in stations that during the examined period were clearly affected from the STT event exhibiting a distinct increase of ozone in the upper-middle troposphere. To our knowledge (visual inspection of ozonesonde data from European stations of the WOUDC network), from the available ozonesonde stations only the station at Prague exhibited a clear ozone enhancement in the middle-upper troposphere. Still, we present below the

ozonesonde data from other European stations (same as Figure 10) during the examined period in support of the models general performance.

[Figure]

Hohenpeissenberg (DE) 06Z 04JAN2017    De Bilt (NL) 12Z 05JAN2017

[Figure]

Legionowo (PL) 12Z 04JAN2017

3. More description is required on the data assimilation. What kinds of observation data were used in the assimilation? Were the ozonesonde data at Prague or aircraft data at Frankfurt used? If so, this should be pointed out when discussing Figures 10 and 11.

The ozonesonde and aircraft data at Prague and Frankfurt respectively were not used in the assimilation process and are therefore completely independent validation data. We have included the following sentences in the revised manuscript:

Page 3, lines 25-28: *"For ozone the CAMS near real time system only assimilates satellite retrievals. These include total column ozone retrievals from the Ozone Monitoring Instrument (OMI) and the Global Ozone Monitoring Experiment-2 (GOME-2) on Metop-A and Metop-B, profile data from the Microwave Limb Sounder (MLS) and partial columns from Solar Backscatter Ultra-Violet (SBUV/2) and from the Ozone Mapping and Profiler Suite (OMPS)."*

Page 5, lines 27-28: *"It is noteworthy to mention that both ozonesondes and IAGOS profiles are not assimilated and hence they constitute completely independent validation data."*

Specific
1. Please indicate the locations of radiosond, ozonesonde stations, and Frankfun in Figures 1-4.

We understand the rationale behind the comment and we thank the Reviewer for the constructive suggestion. We have included the names of the observational sites at their locations in Figures 1-4 (pages 20, 21, 22 and 23 in the revised manuscript) and modified the respective captions accordingly.

2. Figure 10, humidity data usually are available together with the ozonesonde data. Humidity can be validated at the same time.

Please refer to our response in a previous comment. Relative humidity is now included in Figures 5, 6, 7, 9 and 10 in the revised manuscript.

3. References:
P12, L3, CO 2 and CH 4?

Corrected (page 13, lines 3-4 in the revised manuscript).

P12, L13, more information is required.

Done (page 13, line 13 in the revised manuscript).

P12,34, CATHALA?

Corrected (page 13, lines 34-35 in the revised manuscript).

P15, L22, Spell out the full name of the journal.

Done (page 16, line 27 in the revised manuscript).

\*\*\* Page 17, line 8 in the revised manuscript, *"LOTOS?EUROS"* is replaced with *"LOTOS-EUROS"*.

\*\*\* In the revised manuscript for the RegEns ozone vertical profiles the altitude of the sites was considered (if needed) for choosing Standard Atmosphere pressure and temperature.

\*\*\* In the revised manuscript in Figures 11a and c the RegEns ozone is plotted for time 12Z (in order to be consistent with IFS) instead of 13Z that was by mistake in the initial manuscript.

---

## Author Comment (AC2) · 30 Sep 2018

Note: Reviewer's comments are presented in black font; authors' responses are presented in blue plain font; manuscript text quotations are presented in blue italics font.

Anonymous Referee #2

We would like to thank Reviewer #2 for his/her time devoted and the constructive and helpful comments.

Akritidis et al present a study which assesses a stratosphere-to-troposphere transport event (STT) that occurred over Europe during the cold winter of 2017. STT is a very important source of ozone into the troposphere but has and remains challenging to simulate given the laminar like structures that are associated with these events and their transient in time nature. Arkitidis use results from a range of models including the ECMWF Copernicus Atmosphere Monitoring Service to understand the drivers for this event and to use the observations of this event from aircraft and ozonesondes to evaluate the models. In general this is a well written manuscript and one I would recommend published after the following general and technical points are considered.

We thank the Reviewer for the comments, to which we will respond point by point.

General comments:
1. A table with the model acronyms and set ups used in the analysis would be very useful for the reader. Most of the information is already in the text but I feel a table would help the reader quickly appreciate the differences between RegEns, CAMS and IFS.

We agree with the reviewer's suggestion. The CAMS models and simulations used in the current study are now presented and described in Table 1 which is included in the revised manuscript (page 19).

*Table 1. CAMS models and simulations used in the present study.*

| CAMS Models and Simulations | Description |
|---|---|
| CAMS | Copernicus Atmosphere Monitoring Service |
| IFS (CAMS global) | ECMWF Integrated Forecasting System |
| RegEns (CAMS regional ensemble) | Median ensemble of the seven CAMS regional model forecasts |
| IFS Forecast Day-1 | IFS forecast one day in advance |
| IFS Forecast Day-2 | IFS forecast two days in advance |
| IFS Forecast Day-3 | IFS forecast three days in advance |
| IFS Forecast Day-4 | IFS forecast four days in advance |
| IFS Forecast Day-5 | IFS forecast five days in advance |
| IFS no DA Forecast Day-1 | IFS forecast one day in advance without the use of data assimilation |
| RegEns Forecast Day-1 | Regional Ensemble forecast one day in advance |

The following phrase is also included in the revised manuscript, page 4, lines 30-31: *"Table 1 presents the CAMS models and simulations used in the present study."*

2. What more can we learn from this event? The RegEns models all differ in structure and I wonder what further analysis could be done to help understand (a) the role of resolution in the vertical (b) the role of horizontal resolution in biases that occur in the models. The use of the IFS with and without DA is instructive but I feel there is more to be teased out from RegEns and would like to see some more effort to that extent.

We thank the Reviewer for the comment. The role of resolution in vertical would be an interesting sensitivity study, which could provide a concrete answer to the question of how much can we improve a forecast, when we improve the vertical model resolution. If this was an in-house model, we could perform a series of sensitivity tests to investigate the hypothesis. Within CAMS84, however, we have access only to the operational forecast, and our task is to evaluate, as is. The same applies for the regional model forecasts.
In the same way, the impact of horizontal resolution on model performance would be another valuable exercise, which is usually performed with stand-alone models, however, is impossible to perform with operational datasets, with zero flexibility in the model data streams. As evaluation team within CAMS84, we cannot decide on the type of evaluation experiments there should be performed with the CAMS models. We should also note that despite the different regional model set-ups, we have access only to the post-processed datasets, which all have the same horizontal resolution as a result of re-gridding. Regional fields are only available in pre-defined vertical levels (0, 50, 250, 500, 1000, 2000, 3000 and 5000 m).

Technical comments:
Page 1, line 1: Technical point. I'm not sure I agree with the opening statement. STT tends to produce about 200 Tg (O3)/yr. Lightning NOx (natural) produces XXX..

We agree with Reviewer's comment. We have replaced the phrase "*is the dominant natural source of* " with the phrase *"is an important natural source of "* (page 1, line 1 in the revised manuscript).

Page 3, line 24: I find the phrase "weather-chemistry feedback" a bit puzzling.. I think you could be more specific here. What exact feedbacks are included and how are they represented?

More information on the included weather-chemistry feedbacks can be found in Inness et al., 2015, and references therein. In more detail Inness et al. state: *"It was therefore decided to implement the chemistry scheme and its solvers directly in the IFS, together with modules for photolysis, wet and dry deposition, as well as emission injection, to create a more efficient model system called the Composition-IFS (C-IFS, Flemming et al., 2015)".* In the revised manuscript we have included the reference *"(Inness et al., 2015, and references therein)*" after the respective phrase (page 3, line 25).

Page 4, line 2: I guess when you mean data assimilation you mean chemical data

assimilation?

Yes. We have replaced the phrase "data assimilation" with the phrase *"chemical data assimilation"* (page 4, line 7 in the revised manuscript).

Page 4, line 8-9: Pernickety, I know, but you have used "seven" on line 8 and "7" on line 9 to refer to the number of models in the CAMS ensemble. Sticking with one or the other would be better.

Done. We have replaced "7" with "seven" (page 4, line 14 in the revised manuscript).

Page 6, line 10: Define FYROM please.
Done. We have included the definition *"(Former Yugoslav Republic of Macedonia)"* as suggested (page 6, lines 22-23 in the revised manuscript).

Page 6, line 13: Remove "the" before sea-level.
Done (page 6, line 25 in the revised manuscript).

Page 7, line 7: Can the authors confirm why pressure-interpolation of the chemical fields from model levels onto pressure levels could not be performed? I would think this is a fairly standard procedure. Could they elaborate on the errors introduced for example by not accounting for the exceptionally low real temperatures when using the US Standard Atmosphere for unit conversion? In addition, with respect to Figure 4, I would be intrigued to know what the spread is within RegEns or the standard deviation of the ensemble? In general, what can we learn more about the models from this event?

We agree with the Reviewer that the model-to-pressure levels interpolation is a standard procedure. Nevertheless, this study is performed within the framework of CAMS84 where the regional models are strictly provided in 8 height levels (up to 5 km), so actually data in model levels for the regional models are not available.
Regarding the error induced by the use of Standard Atmosphere temperature at 5000m (550.65K and 540.19 hPa) compared to the real temperatures during the period of interest, for a region of lon:10-30E and lat:32-58N and during the time span of Figure 4 there is a decrease of ozone concentrations of about 2.3 ppb when we consider the temperatures from the ERA-Interim dataset at 550 hPa. The spatial distribution of the percentage (%) differences between ozone concentrations calculated using the ERA-Interim temperatures and that of the Standard Atmosphere over the abovementioned time period are shown below. As expected the larger discrepancies in calculated ozone (up to 7%) are found over the regions exhibiting the lower temperatures. The respective text has been modified in the revised manuscript (page 7, lines 18-24) as follows:
*"Although the spatio-temporal features of ozone in the RegEns agree well with that of the IFS, in quantitative terms there are discrepancies between the regional and the global product. This is likely due to the fact that (a) the RegEns is presented at 5000m level (the uppermost level available) and the IFS at 500 hPa, (b) different resolution and advection schemes are used in global and regional models and (c) pressure and temperature values from US Standard Atmosphere (USAF, 1976) were used for units conversion in RegEns. Considering the ERA-Interim (Dee et al., 2011) temperatures during the period of interest for the units conversion may result in even*

*lower RegEns ozone concentrations of up to ~7% in the regions exhibiting the lower temperatures (not shown)."*

The reference Dee et al., 2011 is also included in the revised manuscript (page 14, lines 20-22).

[Figure]

*Percentage (%) differences of ozone concentrations calculated using the ERA-Interim temperatures for the time period 12Z04JAN2017-12Z08JAN2017 and Standard Atmosphere temperatures at 5000m.*

As concerns the spread among the regional models, the following figure shows the standard deviation of the regional ensemble ozone concentrations at 5000m for the same dates as that of Figure 4.

[Figure]

*Standard deviation of the regional ensemble ozone concentrations at 5000m for the time period 12Z04JAN2017-12Z08JAN2017.*

The "take-home message" from this study about the regional models is that the global forcing from the IFS results in capturing an extreme STT event of several stratospheric intrusions over Europe, consistent with the global model.

Page 8, line 16: I'm not sure what you mean by "mind the angle of view"?

We agree with the Reviewer that this is a bit confusing and thus has been removed in the revised manuscript. Our intent was to emphasize the orientation of the 3D structure.

Page 9, line 14: Can you speculate why the spread in RegEns increases in the vertical?

We can assume that the different dynamical cores, the schemes regulating vertical transport, and the methodology to relax top boundary concentrations could lead to the increased spread between the regional ensemble members.

Page 10, line 1: What constitutes "satisfactory"?

We understand that "satisfactory" gives a more qualitative interpretation. Still, a satisfactory forecast performance stands for a model forecast that captures the observed ozone variations in the vertical with a relatively small bias (FGE < 0.3).

Page 10, line 22: Insert "the" before CAMS.

Done (page 11, line 12 in revised manuscript).

Figure 2 caption, missing information about panels e and f.

We thank the Reviewer for the comment. The missing information are now included in Figure 2 caption in the revised manuscript (page 21).

*** Page 17, line 8 in the revised manuscript, *"LOTOS?EUROS"* is replaced with *"LOTOS-EUROS"*.

*** In the revised manuscript for the RegEns ozone vertical profiles the altitude of the sites was considered (if needed) for choosing Standard Atmosphere pressure and temperature.

*** In the revised manuscript in Figures 11a and c the RegEns ozone is plotted for time 12Z (in order to be consistent with IFS) instead of 13Z that was by mistake in the initial manuscript.